# A deep learning approach for automated scoring of the Rey–Osterrieth complex figure

Nicolas Langer[1,2,3]*, Maurice Weber[4], Bruno Hebling Vieira[1,2,3], Dawid Strzelczyk[1,2,3], Lukas Wolf[1], Andreas Pedroni[1,2,3], Jonathan Heitz[1,4], Stephan Müller[4], Christoph Schultheiss[4], Marius Troendle[1,2,3], Juan Carlos Arango Lasprilla[5], Diego Rivera[6,7], Federica Scarpina[8,9], Qianhua Zhao[10], Rico Leuthold[11], Flavia Wehrle[12], Oskar Jenni[12], Peter Brugger[13], Tino Zaehle[14], Romy Lorenz[15,16], Ce Zhang[4]

[1]Methods of Plasticity Research, Department of Psychology, University of Zurich, Zurich, Switzerland; [2]University Research Priority Program (URPP) Dynamics of Healthy Aging, Zurich, Switzerland; [3]Neuroscience Center Zurich (ZNZ), University of Zurich and ETH Zurich, Zurich, Switzerland; [4]Department of Computer Science, ETH Zurich, Zurich, Switzerland; [5]Virginia Commonwealth University, Richmond, United States; [6]Department of Health Science, Public University of Navarre, Pamplona, Spain; [7]Instituto de Investigación Sanitaria de Navarra (IdiSNA), Pamplona, Spain; [8]'Rita Levi Montalcini' Department of Neurosciences, University of Turin, Turin, Italy; [9]IRCCS Istituto Auxologico Italiano, UO di Neurologia e Neuroriabilitazione, Ospedale San Giuseppe, Piancavallo, Italy; [10]Huashan Hospital, Shanghai, China; [11]Smartcode, Zurich, Switzerland; [12]University Children's Hospital Zurich, Child Development Center, Zurich, Switzerland; [13]Rehabilitation Center, Valens, Switzerland; [14]University Hospital Magdeburg University Department of Neurology, Magdeburg, Germany; [15]Max Planck Institute for Biological Cybernetics, Tübingen, Germany; [16]Max Planck Institute for Human Cognitive and Brain Sciences, Leipzig, Germany

*For correspondence:
n.langer@psychologie.uzh.ch

Competing interest: The authors declare that no competing interests exist.

## eLife assessment

The methods and findings of the current work are **important** and well-grounded. The strength of the evidence presented is **convincing** and backed up by rigorous methodology. The work, when elaborated on how to access the app, will have far-reaching implications for current clinical practice.

**Abstract** Memory deficits are a hallmark of many different neurological and psychiatric conditions. The Rey–Osterrieth complex figure (ROCF) is the state-of-the-art assessment tool for neuropsychologists across the globe to assess the degree of non-verbal visual memory deterioration. To obtain a score, a trained clinician inspects a patient's ROCF drawing and quantifies deviations from the original figure. This manual procedure is time-consuming, slow and scores vary depending on the clinician's experience, motivation, and tiredness. Here, we leverage novel deep learning architectures to automatize the rating of memory deficits. For this, we collected more than 20k hand-drawn ROCF drawings from patients with various neurological and psychiatric disorders as well as healthy participants. Unbiased ground truth ROCF scores were obtained from crowdsourced human intelligence. This dataset was used to train and evaluate a multihead convolutional neural network.

The model performs highly unbiased as it yielded predictions very close to the ground truth and the error was similarly distributed around zero. The neural network outperforms both online raters and clinicians. The scoring system can reliably identify and accurately score individual figure elements in previously unseen ROCF drawings, which facilitates explainability of the AI-scoring system. To ensure generalizability and clinical utility, the model performance was successfully replicated in a large independent prospective validation study that was pre-registered prior to data collection. Our AI-powered scoring system provides healthcare institutions worldwide with a digital tool to assess objectively, reliably, and time-efficiently the performance in the ROCF test from hand-drawn images.

## Introduction

Neurological and psychiatric disorders are among the most common and debilitating illnesses across the lifespan. In addition, the aging of our population, with the increasing prevalence of physical and cognitive disorders, poses a major burden on our society with an estimated economic cost of 2.5 trillion US$ per year (*Trautmann et al., 2016*). Currently, neuropsychologists typically use paper-pencil tests to assess individual neuropsychological functions and brain dysfunctions, including memory, attention, reasoning, and problem-solving. Most neuropsychologists around the world use the Rey–Osterrieth complex figure (ROCF) in their daily clinical practice (*Rabin et al., 2005*; *Rabin et al., 2016*), which provides insights into a person's non-verbal visuo-spatial memory capacity in healthy and clinical populations of all ages, from childhood to old age (*Shin et al., 2006*).

Our estimation revealed that a single neuropsychological division (e.g. at the University Hospital Zurich) scores up to 6000 ROCF drawings per year. The ROCF test has several advantages as it does not depend on auditory processing and differences in language skills that are omnipresent in cosmopolitan societies (*Osterrieth, 1944*; *Somerville et al., 2000*). The test has adequate psychometric properties (e.g. sufficient test–retest reliability [*Meyers and Volbrecht, 1999*; *Levine et al., 2004*] and internal consistency [*Berry et al., 1991*; *Fastenau et al., 1996*]). Furthermore, the ROCF test has demonstrated to be sensitive to discriminate between various clinical populations (*Alladi et al., 2006*) and the progression of Alzheimer's disease (*Trojano and Gainotti, 2016*).

The ROCF test consists of three test conditions: First, in the *Copy condition* subjects are presented with the ROCF and are asked to draw a copy of the same figure. Subsequently, the ROCF figure and the drawn copy are removed and the subject is instructed to reproduce the figure from memory immediately (*Shin et al., 2006*) or 3 min after the Copy condition (*Meyers et al., 1996*) (*Immediate Recall condition*). After a delay of 30 min, the subject is required to draw the same figure once again (*Delayed Recall* condition). For further description of the clinical usefulness of the ROCF please refer to *Shin et al., 2006*.

The current quantitative scoring system (*Meyers et al., 1996*) splits the ROCF into 18 identifiable elements (see *Figure 1A*), each of which is considered separately and marked on the accuracy in both distortion and placement according to a set of rules and scored between 0 and 2 points (see *Figure 1—figure supplement 1*). Thus, the scoring is currently undertaken manually by a trained clinician, who inspects the reproduced ROCF drawing and tracks deviations from the original figure, which can take up to 15 min per figure (individually *Copy*, *Immediate Recall*, and *Delayed Recall* conditions).

One major limitation of this quantitative scoring system is that the criteria of what position and distortion is considered 'accurate' or 'inaccurate' may vary from clinician to clinician (*Groth-Marnat, 2000*; *Shin et al., 2006*; *Canham et al., 2000*). In addition, the scoring might vary as a function of motivation and tiredness or because the clinicians may be unwittingly influenced by interaction with the patient. Therefore, an automated system that offers reliable, objective, robust, and standardized scoring, while saving clinicians' time, would be desirable from an economic perspective and more importantly leads to more accurate scoring and subsequently diagnosing.

In recent years, computerized image analysis and machine-learning methods have entered the clinical neuropsychological diagnosis field providing the potential for establishing quantitative scoring methods. Using machine-learning methods, such as convolutional neural networks and support vector machines, studies have successfully recognized visual structures of interest produced by subjects in the Bender Gestalt Test (*Bin Nazar et al., 2017*) and the Clock Draw Task (*Kim et al., 2011*; *Harbi et al., 2016*) – both are less frequently applied screening tests for visuo-spatial and visuo-constructive (dis-)abilities (*Webb et al., 2021*). Given the wide application of the ROCF, it is not surprising that

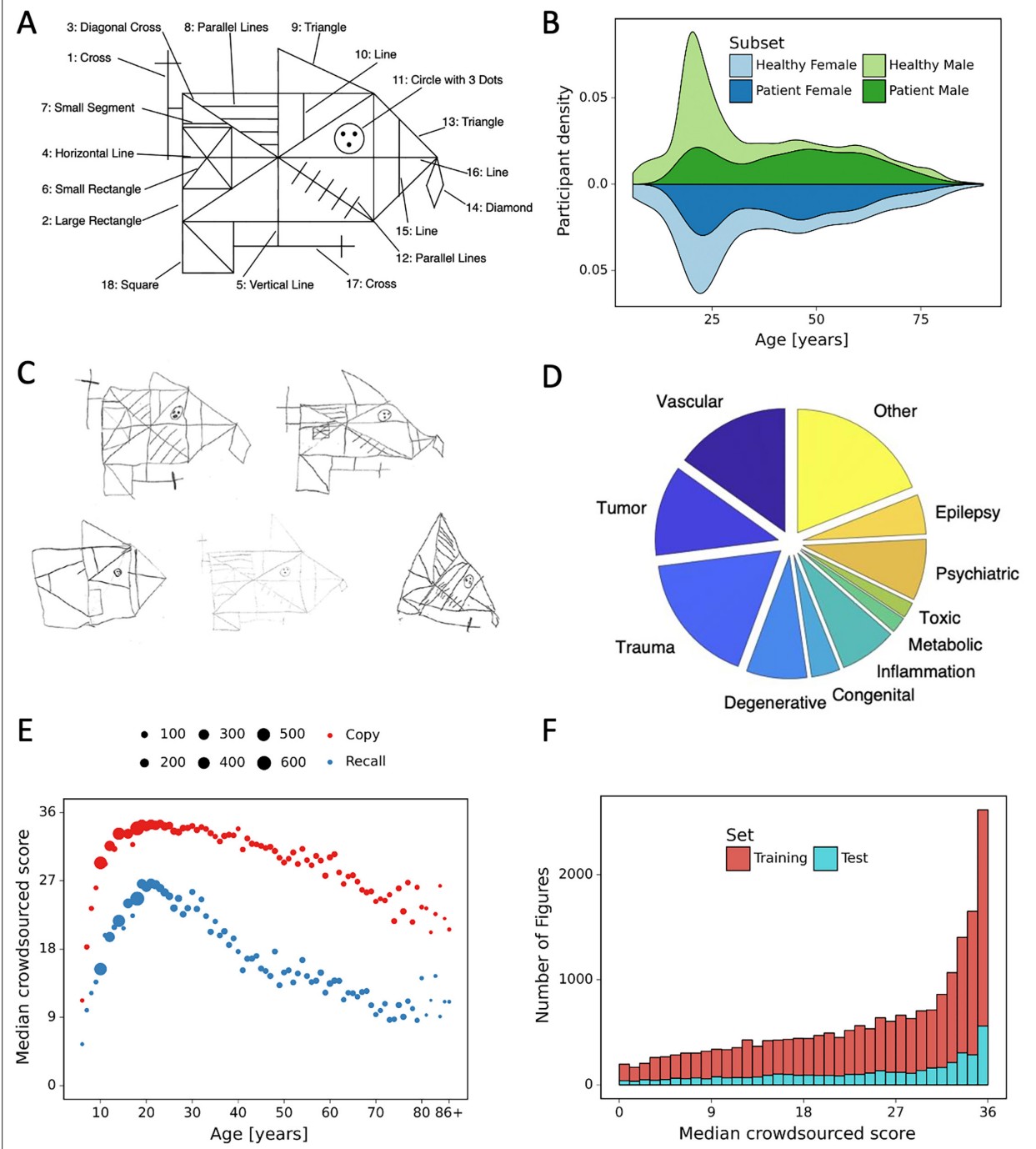

**Figure 1.** Overview of retrospective dataset. (**A**) Rey–Osterrieth complex figure (ROCF) figure with 18 elements. (**B**) Demographics of the participants and clinical population of the retrospective dataset. (**C**) Examples of hand-drawn ROCF images. (**D**) The pie chart illustrates the proportion of the different clinical conditions of the retrospective dataset. (**E**) Performance in the copy and (immediate) recall condition across the lifespan in the retrospective dataset. (**F**) Distribution of the number of images for each total score (online raters).

The online version of this article includes the following figure supplement(s) for figure 1:

**Figure supplement 1.** Original scoring system according to Osterrieth.

**Figure supplement 2.** World maps depict the worldwide distribution of the origin of the data.

**Figure supplement 3.** The graphical user interface of the crowdsourcing application.

**Figure supplement 4.** Overview of prospective dataset.

**Figure supplement 5.** The user interface for the tablet- (and smartphone-) based application.

we are not the first to take steps toward a machine-based scoring system. *Canham et al., 2000* have developed an algorithm to automatically identify a selection of parts of the ROCF (6 of 18 elements) with great precision. This approach provides first evidence for the feasibility of automated segmentation and feature recognition of the ROCF. More recently, *Vogt et al., 2019* developed an automated scoring using a deep neural network. The authors reported a *r* = 0.88 Pearson correlation with human ratings, but equivalence testing demonstrated that the automated scoring did not produce strictly equivalent total scores compared to the human ratings. Moreover, it is unclear if the reported correlation was observed in an independent test dataset, or in the training set. Finally, *Petilli et al., 2021* have proposed a novel tablet-based digital system for the assessment of the ROCF task, which provides the opportunity to extract a variety of parameters such as spatial, procedural, and kinematic scores. However, none of the studies described have been able to produce machine-based scores according to the original scoring system currently used in clinics (*Meyers et al., 1996*) that are equivalent or superior to human raters. A major challenge in developing automated scoring systems is to gather a rich enough set of data with instances of all possible mistakes that humans can make. In this study, we have trained convolutional neural networks with over 20,000 digitized ROCFs from different populations regarding age and diagnostic status (healthy individuals or individuals with neurological and psychiatric disorders [e.g. Alzheimer, Parkinson]).

## Results

### A human MSE and clinicians' scoring

We have harnessed a large pool (~5000) of human workers (crowdsourced human intelligence) to obtain ground truth scores and compute the *human MSE*. An average of 13.38 (sd = 2.23) crowdsource-based scores per figure was gathered. The *average human MSE* over all images is 16.3, and the *average human MAE* is 2.41.

For a subset (4030) of the 20,225 ROCF images, we had access to scores conducted by different professional clinicians. This enabled us to analyze and compare the scoring of professionals to the crowdsourcing results. The clinician MSE over all images is 9.25 and the clinician MAE is 2.15, indicating a better performance of the clinicians compared to the average human rater.

### Machine-based scoring system

The multilabel classification network achieved a total score mean squared error (MSE) of 3.56 and mean absolute error (MAE) of 1.22, which is already considerably lower than the corresponding human performance. Implementing additional data augmentation (DA) has led to a further improvement in accuracy with an MSE of 3.37 and MAE of 1.16. Furthermore, when combining DA with test-time augmentation (TTA) leads to a further improvement in performance with an MSE of 3.32 and MAE of 1.15.

The regression-based network variant led to a further slight improvement in performance, reducing the total score MSE to 3.07 and the MAE to 1.14. Finally, our best model results from combining the regression model with the multilabel classification model in the following way: For each item of the figure we determine whether to use the regressor or classifier based on its performance on the validation set (*Figure 2B*). Aggregating the two models in this way leads to an MSE of 3.00 and an MAE of 1.11. *Figure 2C* presents the error for all combinations of DA and TTA with the multilabel classification and regression model separately and in combination. During the experiments, it became apparent that for the multilabel classification network applying both DA and TTA jointly improves the model's performance (see *Figure 2C*). Somewhat surprisingly, applying these elements to the multi-head regression model did not improve the performance compared to the non-augmented version of the model. The exact performance metrics (MSE, MAE, and *R*-squared) of all model variants are reported in *Figure 2—source data 1* and for each figure element in *Figure 2—source data 2*. In addition, the model performance was replicated in the independent prospective validation study (i.e. MSE = 3.32; MAE = 1.13). The performance metrics of each figure element for the independent prospective validation study are reported in *Figure 2—source data 3*.

We further conducted a more fine-grained analysis of our best model. *Figure 3A* shows the score for each figure in the dataset contrasted with the ground truth score (i.e. median of online raters). In addition, we computed the difference between the ground truth score and predicted score revealing

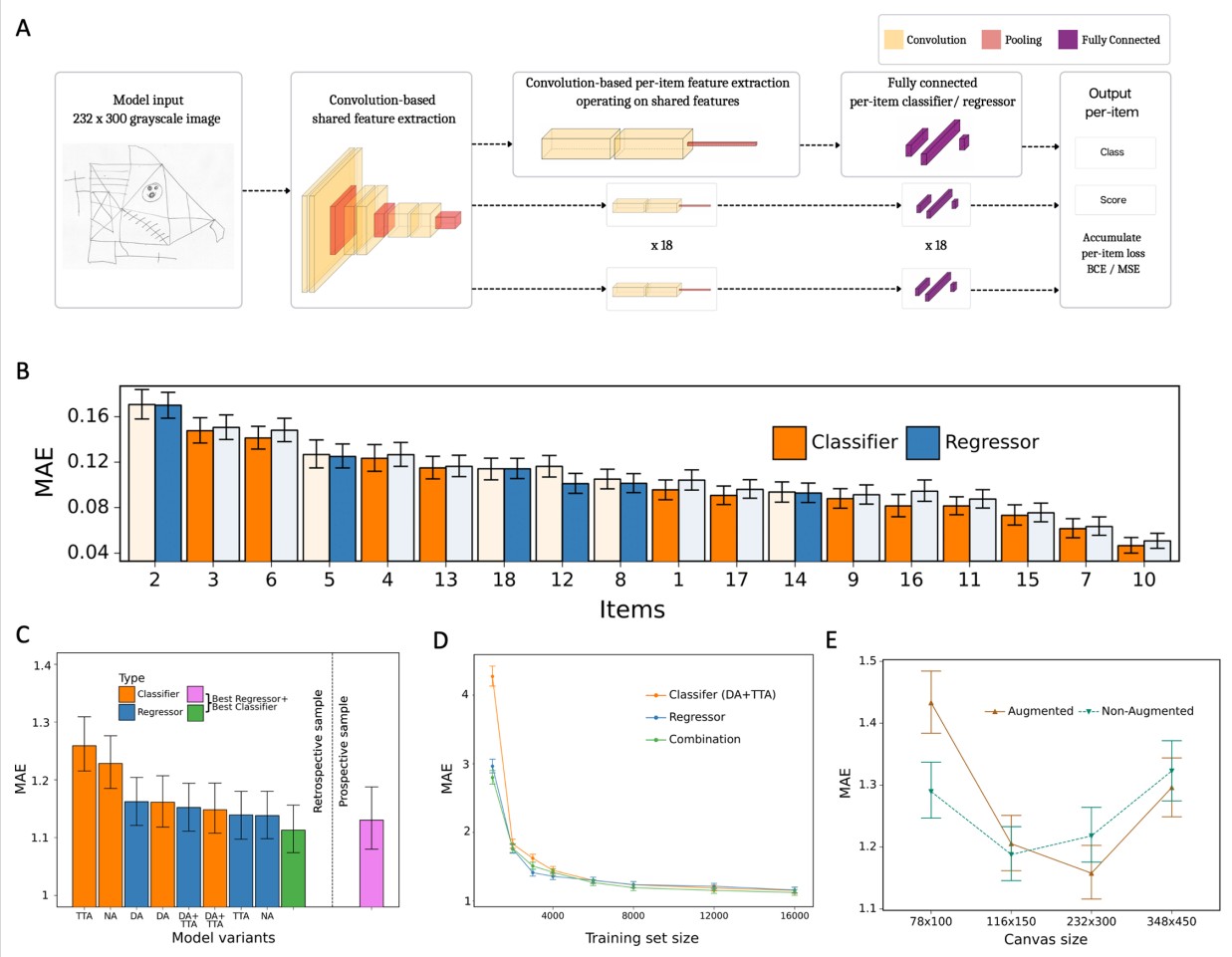

**Figure 2.** Model architecture and performance evaluation. (**A**) Network architecture, constituted of a shared feature extractor and 18 item-specific feature extractors and output blocks. The shared feature extractor consists of three convolutional blocks, whereas item-specific feature extractors have one convolutional block with global max pooling. Convolutional blocks consist of two convolution and batch normalization pairs, followed by max pooling. Output blocks consist of two fully connected layers. ReLU activation is applied after batch normalization. After pooling, dropout is applied. (**B**) Item-specific mean absolute error (MAE) for the regression-based network (blue) and multilabel classification network (orange). In the final model, we determine whether to use the regressor or classifier network based on its performance in the validation dataset, indicated by an opaque color in the bar chart. In case of identical performance, the model resulting in the least variance was selected. (**C**) Model variants were compared and the performance of the best model in the original, retrospectively collected (green) and the independent, prospectively collected (purple) test set is displayed; Clf: multilabel classification network; Reg: regression-based network; NA: no augmentation; DA: data augmentation; TTA: test-time augmentation. (**D**) Convergence analysis revealed that after ~8000 images, no substantial improvements could be achieved by including more data. (**E**) The effect of image size on the model performance is measured in terms of MAE. The error bars in all subplots indicate the 95% confidence interval.

The online version of this article includes the following source data for figure 2:

**Source data 1.** The performance metrics for all model variants.

**Source data 2.** Per-item and total performance estimates for the final model of the retrospective data.

**Source data 3.** Per-item and total performance estimates for the final model with prospective data.

that errors made by our model are concentrated closely around 0 (*Figure 3B*), while the distribution of clinician's errors is much more spread out (*Figure 3D, E*). It is interesting to note that, like the clinicians, our model exhibits a slight bias toward higher scores, although much less pronounced. Importantly, the model does not demonstrate any considerable bias toward specific figure elements. In contrast to the clinicians, the MAE is very balanced across each individual item of the figure (*Figure 3C, F* and *Figure 2—source data 2*). Finally, a detailed breakdown of the expected performance across the entire range of total scores is displayed for the model (*Figure 3G* and *Figure 3—source data 1*), the clinicians (*Figure 3H*), and average online raters (*Figure 3I*). As can be seen, the model exhibits

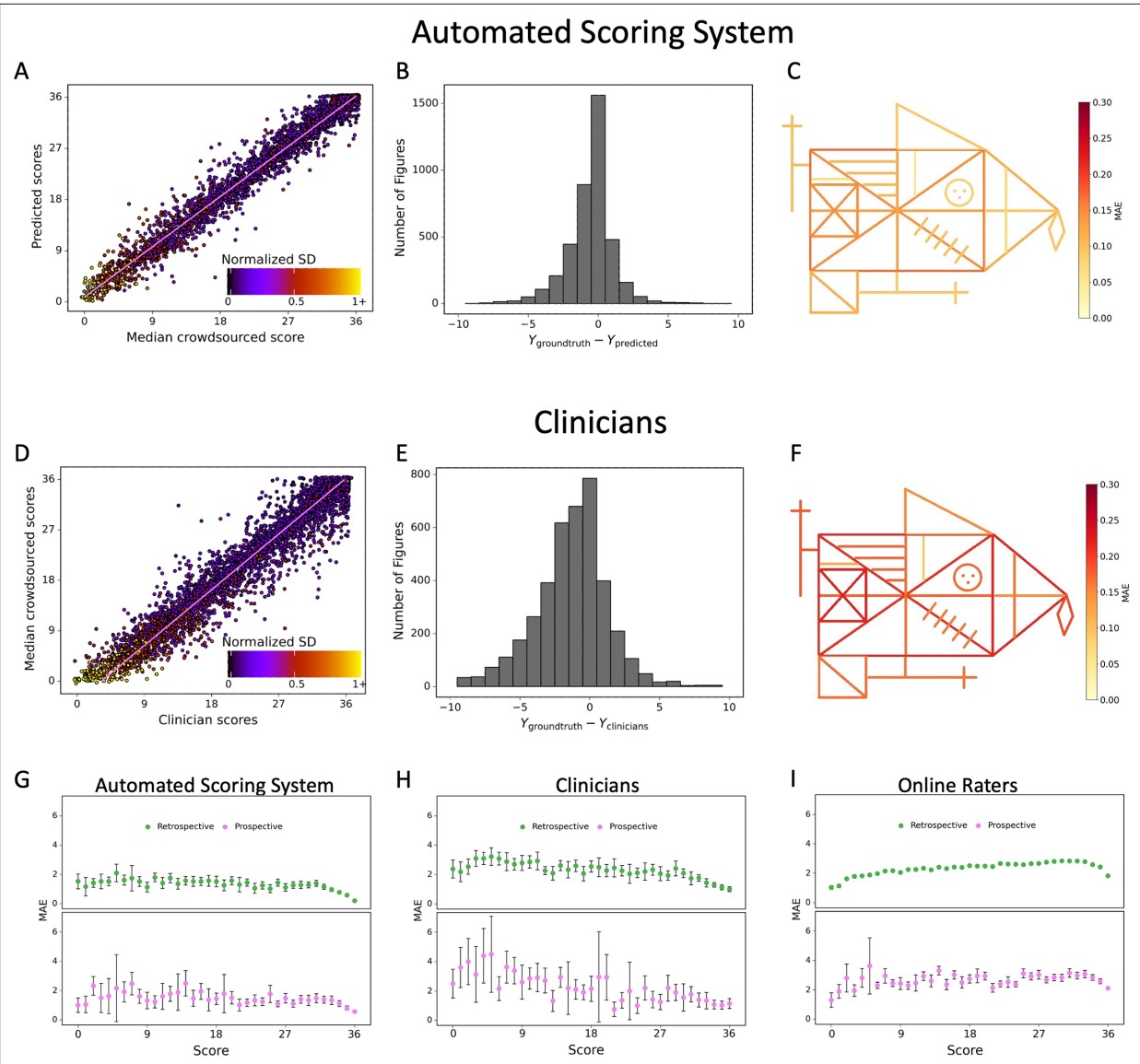

**Figure 3.** Contrasting the ratings of our model (**A**) and clinicians (**D**) against the ground truth revealed a larger deviation from the regression line for the clinicians. A jitter is applied to better highlight the dot density. The distribution of errors for our model (**B**) and the clinicians ratings (**E**) is displayed. The mean absolute error (MAE) of our model (**C**) and the clinicians (**F**) is displayed for each individual item of the figure (see also ***Figure 2—source data 1***). The corresponding plots for the performance on the prospectively collected data are displayed in ***Figure 3—figure supplement 1***. The model performance for the retrospective (green) and prospective (purple) sample across the entire range of total scores for model (**G**), clinicians (**H**), and online raters (**I**) is presented. The error bars in all subplots indicate the 95% confidence interval.

The online version of this article includes the following source data and figure supplement(s) for figure 3:

**Source data 1.** Performance per total score interval with retrospective data.

**Source data 2.** Performance per total score interval with prospective data.

**Figure supplement 1.** Detailed performance of the model on the prospective data.

**Figure supplement 2.** The standard deviation of the human raters is displayed across differently scored drawings.

a comparable MAE for the entire range of total scores, although there is a trend that high scores exhibit lower MAEs. These results were confirmed in the independent prospective validation study (see ***Figure 3G–I***, ***Figure 3—figure supplement 1***, and ***Figure 3—source data 2***).

In addition, we have conducted a comprehensive model performance analysis to evaluate our model's performance across different ROCF conditions (copy and recall), demographics (age and gender), and clinical statuses (healthy individuals and patients) (***Figure 4A***). These results have been

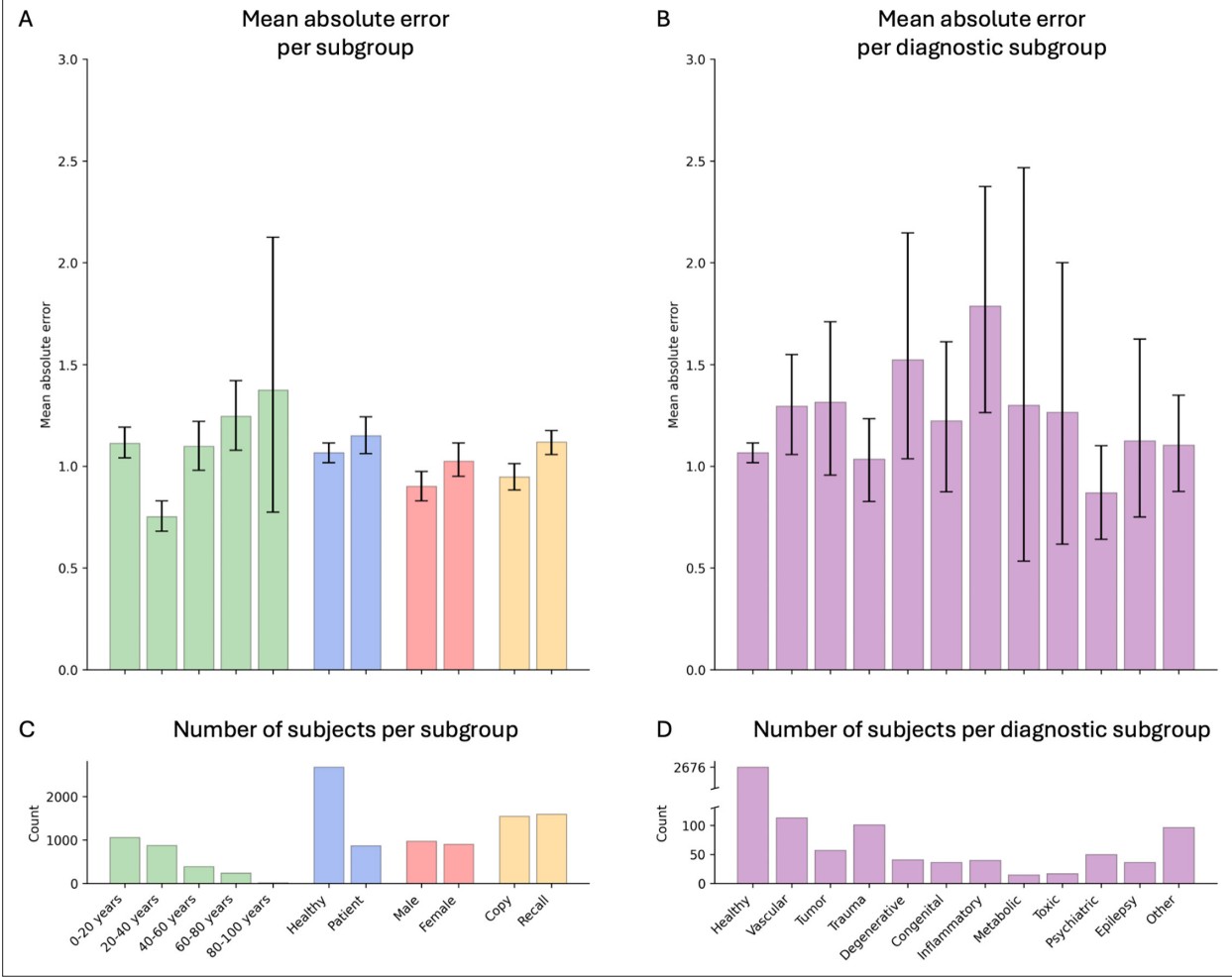

**Figure 4.** Model performance across ROCF conditions, demographics, and clinical subgroups in the retrospective dataset. (**A**) Displayed are the mean absolute error and bootstrapped 95% confidence intervals of the model performance across different Rey–Osterrieth complex figure (ROCF) conditions (copy and recall), demographics (age and gender), and clinical statuses (healthy individuals and patients) for the retrospective data. (**B**) Model performance across different diagnostic conditions. (**C, D**) The number of subjects in each subgroup is depicted. The same model performance analysis for the prospective data is reported in *Figure 4—figure supplement 1*.

The online version of this article includes the following figure supplement(s) for figure 4:

**Figure supplement 1.** Model performance across ROCF conditions, demographics, and clinical subgroups in prospective dataset.

confirmed in the prospective validation study (*Figure 4—figure supplement 1*). Furthermore, we included an additional analysis focusing on specific diagnoses to assess the model's performance in diverse patient populations (*Figure 4B*). Our findings demonstrate that the model maintains high accuracy and generalizes well across various demographics and clinical conditions.

## Robustness analysis

Using DA did not critically improve the accuracy, which is likely due to the fact that our dataset is already large and diverse enough. However, DA does significantly improve robustness against semantic transformations, as can be seen from *Figure 5—figure supplement 1*. In particular, using our DA pipeline, the model becomes much more robust against rotations and changes in perspective. On the other hand, for changes to brightness and contrast, we could not find a clear trend. This is however not surprising as the DA pipeline does not explicitly encourage the model to be more robust against these transformations. Overall, we demonstrate that our scoring system is highly robust for realistic changes in rotations, perspective, brightness, and contrast of the images (*Figure 5*). Performance is degraded only under unrealistic and extreme transformations.

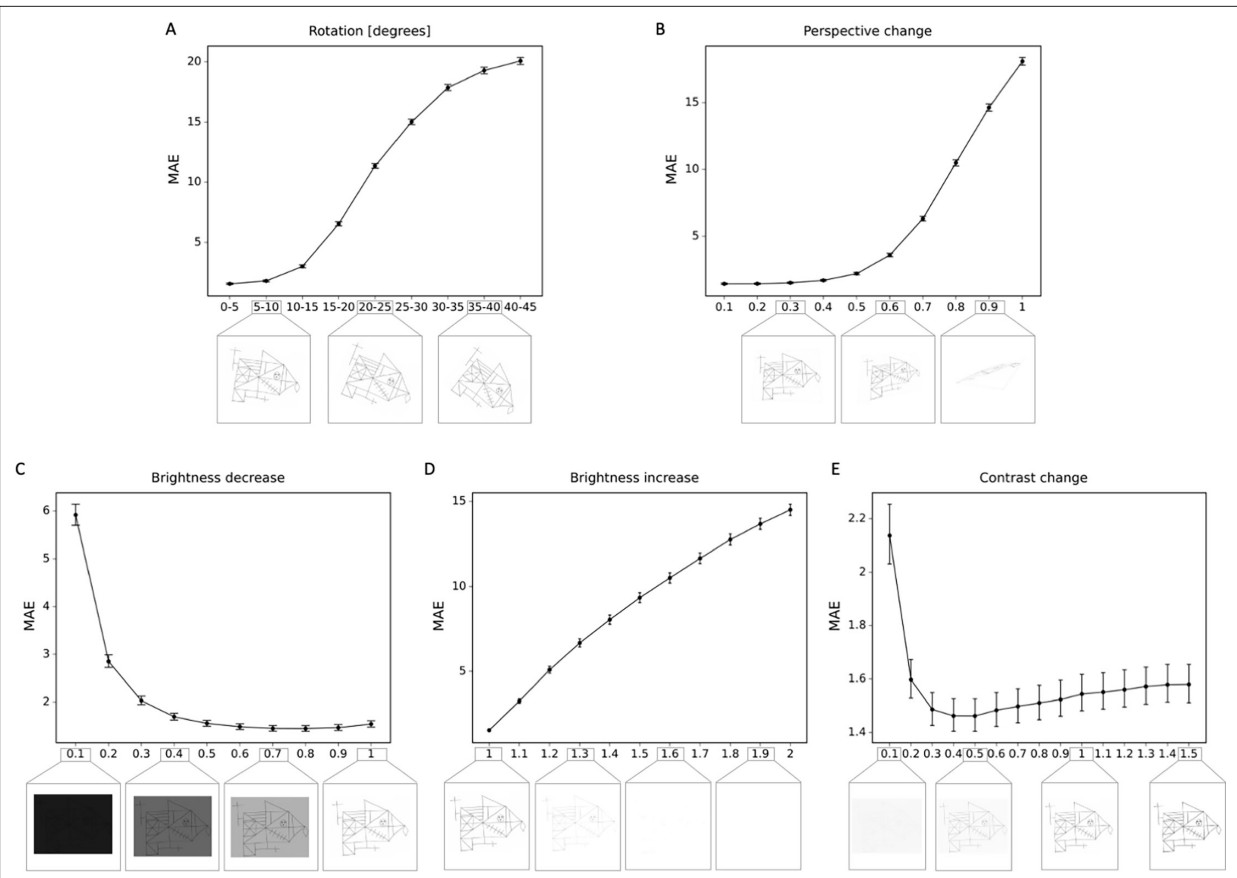

**Figure 5.** Robustness to geometric, brightness, and contrast variations. The mean absolute error (MAE) is depicted for different degrees of transformations, including (**A**) rotations; (**B**) perspective change; (**C**) brightness decrease; (**D**) brightness increase; (**E**) contrast change. In addition examples of the transformed Rey–Osterrieth complex figure (ROCF) draw are provided. The error bars in all subplots indicate the 95% confidence interval.

The online version of this article includes the following figure supplement(s) for figure 5:

**Figure supplement 1.** Effect of data augmentation.

## Discussion

In this study, we developed an AI-based scoring system for a non-verbal visuo-spatial memory test that is being utilized in clinics around the world on a daily basis. For this, we trained two variations of deep learning systems and used a rich dataset of more than 20k hand-drawn ROCF images covering the entire lifespan, different research and clinical environments as well as representing global diversity. By leveraging human crowdsourcing we obtained unbiased high-precision training labels. Our best model results from combining a multihead convolutional neural network for regression with a network for multilabel classification. DA and TTA were used to improve the accuracy and made our model more robust against geometric transformations like rotations and changes in perspective.

Overall, our results provide evidence that our AI-scoring tool outperforms both amateur raters and clinicians. Our model performed highly unbiased as it yielded predictions very close to the ground truth and the error was similarly distributed around zero. Importantly, these results have been replicated in an independent prospectively collected dataset. The scoring system reliably identified and accurately scored individual figure elements in previously unseen ROCF drawings, which facilitates the explainability of the AI-scoring system. Comparable performance for each figure element indicates no considerable bias toward specific figure compositions. Furthermore, the error of our scoring system is rather balanced across the entire range of total scores. An exception are the highest scored images displaying the smallest error, which has two explanations: First, compared to low-score images, for which the same score can be achieved by numerous combinations of figure elements, the high-score

images by nature do not have as many possible combinations. Second, the dataset contains a proportional larger amount of available training data for the high-score images.

While the ROCF is one of the most commonly employed neuropsychological test to evaluate non-verbal visuo-spatial memory in clinical setting (*Shin et al., 2006*), the current manual scoring in clinics is time-consuming (5–15 min for each drawing), requires training, and ultimately depends on subjective judgments, which inevitably introduces human scoring biases (*Watkins, 2017*; *Franzen, 2000*) and low inter-rater reliability (*Franzen, 2000*; *Huygelier et al., 2020*; see also *Figure 3—figure supplement 2*). Thus, an objective and reliable automated scoring system is in great demand as it can overcome the problem of intra- and inter-rater variability. More importantly, such an automated system can remove a time-consuming, tedious, and repetitive task from clinicians and thereby helps to reduce the workload of clinicians and/or allow more time for more patient-centered care. Overall, automation can support clinicians in making healthcare decisions more accurate and timely.

Over the past years, deep learning has made significant headway in achieving above human-level performance on well-specified and isolated tasks in computer vision. However, unleashing the power of deep learning depends on the availability of large training sets with accurate training labels. We obtained over 20k hand-drawn ROCF images and invested immense time and labor to manually scan the ROCF drawings. This effort was only possible by orchestrating across multiple clinical sites and having funds available to hire the workforce to accomplish the digitization. Our approach highlights the need to intensify national and international effort of the digitalization of health record data. Although electronic health records are increasingly available (e.g. quadrupled in the US from 2007 to 2012 [*Hsiao et al., 2014*]), challenges remain in terms of handling, processing, and moving such big data. Improved data accessibility and better consensus in organization and sharing could simplify and foster similar approaches in neuropsychology and medicine.

Another important difficulty is that data typically comes from various sources and thus exhibits large heterogeneity in terms of quality, size, format of the images, and crucially the quality of the labeled dataset. High-quality labeled and annotated datasets are the foundation of a successful computer vision system. A novelty of our approach is that we have trained a large pool of human internet workers (crowdsourced human intelligence) to score ROCFs drawings guided by our self-developed interactive web application. Each element of the figure was scored by several human workers (13 workers on average per figure). To derive ground truth scores, we took the median score, as it has the advantage of being robust to outliers. To further ensure high-quality data annotation, we identified and excluded crowdsourcing participants that have a high level of disagreement (>20% disagreement) with this rating from trained clinicians, who carefully scored manually a subset of the data in the same interactive web application. Importantly our twofold innovative approach that combines the digitization of neuropsychological tests and the high-quality scoring using crowdsourced intelligence can provide a roadmap for how AI can be leveraged in neuropsychological testing more generally as it can be easily adapted and applied to various other neuropsychological tests (e.g. Clock Drawing Test [*Morris, 1994*], Taylor Complex Figure Test [*Awad et al., 2004*], Hamasch 5-point test [*Regard et al., 1982*]).

Another important prerequisite of using AI for the automated scoring of neuropsychological tests is the availability of training data that is sufficiently diverse and obtains sufficient examples from all possible scores. Recent examples in computer vision have demonstrated that insufficient diversity in training data can lead to biases and adverse consequences (*Buolamwini and Gebru, 2018*; *Ensign et al., 2017*). Given that our training dataset included data from children and a representative sample of patients (see *Figure 1B*), exhibiting a range of neurological disorders, drawings were frequently extremely dissimilar to the target figure. Importantly, our scoring system delivers accurate scores even in cases where drawings are distorted (e.g. caused by fine motor impairments) or omissions (e.g. due to visuo-spatial deficits), which are typical impairment patterns in neurological patients. In addition, our scoring system is robust against different semantic transformations (e.g. rotations, perspective change, brightness) which are likely to occur in real-life scenarios, when the examiner takes a photo of the ROCF drawing, which naturally will lead to varying viewpoints and illumination conditions. Thus, this robustness is a pivotal prerequisite for any potential clinical utility.

To improve the usability of our system and guarantee clinical utility, we have integrated our model into a tablet- (and smartphone-) based application, in which users can take a photo (or upload an image) of an ROCF drawing and the application instantly provides a total score. The application is

currently in beta testing with selected clinicians in real-world settings. Once beta testing is complete, the application will be made publicly accessible to clinicians and healthcare institutions worldwide, with detailed access and usage instructions available on our website. Importantly, the automated scoring system also maintains explainability by providing visualization of detailed score breakdowns (i.e. item-specific scores), which is a highly valued property of AI-assisted medical decision making and key to help interpret the score and communicate them to the patient (see *Figure 1—figure supplement 5*). The scoring is completely reproducible and thus facilitates valid large-scale comparisons of ROCF data, which enable population-based cognitive screening and (assisted) self-assessments.

In summary, we have created a highly robust and automated AI-based scoring tool, which provides unbiased, reproducible, and immediate scores for the ROCF test in research and clinical environments. Our current machine-based automated quantitative scoring system outperforms both amateur raters and clinicians.

The present findings demonstrate that automated scoring systems can be developed to advance the quality of neuropsychological assessments by diminishing reliance on subjective ratings and efficiently improving scoring in terms of time and costs. Our innovative approach can be translated to many different neuropsychological tests, thus providing a roadmap for paving the way to AI-guided neuropsychology.

## Materials and methods

### Data

For our experiments, we used a dataset of 20,225 hand-drawn ROCF images collected from 90 different countries (see *Figure 1—figure supplement 2*) as well as various research and clinical settings. This large dataset spans the complete range of ROCF scores (*Figure 1F*) which allows the development of a more robust and generalizable automated scoring system. Our convergence analysis suggests that the error converges when approaching 10,000 digitized ROCFs. The data is collected from various populations regarding age and diagnostic status (healthy or with neurological and/or psychiatric disorder), shown, respectively, in *Figure 1E and D*. The demographics of the participants and some example images are shown, respectively, in *Figure 1B and C*. For a subset of the figures (4030), the clinician's scores were available. For each figure only one clinician rating is available. The clinicians ratings were derived from six different international clinics (University Hospital Zurich, Switzerland; University Children's Hospital Zurich, Switzerland; BioCruces Health Research Institute, Spain; I.R.C.C.S. Istituto Auxologico Italiano, Ospedale San Giuseppe, Italy; Huashan Hospital, China; University Hospital Magdeburg, Germany).

The study was approved by the Institutional Ethics Review Board of the 'Kantonale Ethikkommission' (BASEC-Nr. 2020-00206). All collaborators have written informed consent and/or data usage agreements for the recorded drawings from the participants. The authors assert that all procedures contributing to this work comply with the ethical standards of the relevant national and institutional committees on human experimentation and with the Helsinki Declaration of 1975, as revised in 2008. To ensure generalizability and clinical utility, the model performance was replicated in a large independent prospective validation study that was pre-registered prior to data collection (https://osf.io/82796). For the independent prospective validation study, an additional dataset was collected and contained 2498 ROCF images from 961 healthy adults from and 288 patients with various neurological and psychiatric disorders. Further information about the participants demographics and performance in the copy and recall condition can be found in *Figure 1—figure supplement 4*.

### Convergence analysis

To get an estimate on the number of ROCFs needed to train our models we conducted a convergence analysis. That is, for a fixed test set with 4045 samples, we used different fractions of the remaining dataset to train the multilabel classification model, ending up with training set sizes of 1000, 2000, 3000, 4000, 6000, 8000, 12,000, and 16,000 samples. By evaluating the performance (i.e. MAE) of the resulting models on the fixed test set, we determined what amount of data is required for the model performance to converge.

With ~3000 images, we obtain diminishing mean MAEs. After ~10,000 images, no substantial improvements could be achieved by including more data, as can be seen from the progression plot

in *Figure 2D*. From this, we conclude that the acquired dataset is rich and diverse enough to obtain a powerful deep learning model for the task at hand. In addition, this opens up an avenue for future research in that improvements to the model performance are likely to be achieved via algorithmic improvements, rather than via data-centric efforts.

## Crowdsourced human intelligence for ground truth score

To reach high accuracy in predicting individual sample scores of the ROCFs, it is imperative that the scores of the training set are based on a systematic scheme with as little human bias as possible influencing the score. However, our analysis (see results section) and previous work (*Canham et al., 2000*) suggested that the scoring conducted by clinicians may not be consistent, because the clinicians may be unwittingly influenced by the interaction with the patient/participant or by the clinicians factor (e.g. motivation and fatigue). For this reason, we have harnessed a large pool (~5000) of human workers (crowdsourced human intelligence) who scored ROCFs, guided by our self-developed interactive web application (see *Figure 1—figure supplement 3*). In this web application, the worker was first trained by guided instructions and examples. Subsequently, the worker rated each area of real ROCF drawings separately to guarantee a systematic approach. For each of the 18 elements in the figure, participants were asked to answer three binary questions: Is the item visible and recognizable? Is the item in the right place? Is the item drawn correctly? This corresponds to the original Osterrieth scoring system (*Meyers et al., 1996*; see *Figure 1—figure supplement 1*). The final assessment consisted of answers to these questions for each of the 18 items and enabled the calculation of item-wise scores (possible scores: 0, 0.5, 1, and 2), which enabled to compute the total score for each image (i.e. sum over all 18 item-wise scores: range total score: 0–36).

Since participants are paid according to how many images they score, there is a danger of collecting unreliable scores when participants rush through too quickly. In order to avoid taking such assessments into account, 600 images have also been carefully scored manually by trained clinicians at the neuropsychological unit of the University Hospital Zurich (in the same interactive web application). We ensured that each crowdsourcing worker rated at least two of these 600 images, which resulted in 108 ratings (2 figures * 18 items * 3 questions) to compute a level of disagreement. The assessments of crowdsourcing participants that have a high level of disagreement (>20% disagreement) with this rating from clinicians are considered cheaters and are excluded from the dataset. After this data cleansing, there remained an average of 13.38 crowdsource-based scores per figure. In order to use this information for training an algorithm, we require one ground truth score for each image. We assume that the scores approximately follow a normal distribution centered around the true score. With this in mind, we have obtained the ground truth for each drawing by computing the median for each item in the figure, and then summed up the medians to get the total score for the drawing in question. Taking the median has the advantage of being robust to outliers. The identical crowdsourcing approach has also been used to obtain ground truth scores for the independent prospective validation study (an average of 10.36 crowdsource-based scores per figure; minimum number of ratings = 10).

## Human MSE

As described in the previous section, there are multiple independent scores (from different crowdsourcing workers) available for each image. It frequently happens that two people scoring the same image produce different scores. In order to quantify the disagreement among the crowdsourcing participants, we calculated the empirical standard deviation of the scores for each image. With $s1$, …, $sk$ referring to the $k$ crowdsourcing scores for a given image and $\underline{s}$ to their mean, the empirical standard deviation is calculated as

$$SD_{empirical} = \sqrt{\frac{1}{k-1} \sum_{i=1}^{k} \left( s_i - \bar{s} \right)^2}$$

The mean empirical standard deviation is 3.25. Using as ground truth the sum of item score medians, we can also calculate a *human MSE*. Assuming that the sum of median scores of all items of an image is the ground truth, the average rating performance of human raters can be estimated by computing the MSE and the MAE of the human ratings by first computing the mean error for each

human rater, and then computing the mean of all individual MSEs. These metrics describe how close one assessment is to this ground truth on average and let us make a statement on the difficulty of the task. Reusing above notation, we denote the k crowdsourcing scores for a given image by $s_1, \ldots, s_k$. Let $\tilde{s}$ be their median. For an image or item, we define the *human MSE* as

$$MSE_{human} = \frac{1}{k} \sum_{i=1}^{k} \left( s_i - \tilde{s}_i \right)^2 .$$

Similarly, we define the human MAE as

$$MAE_{human} = \frac{1}{k} \sum_{i=1}^{k} |s_i - \tilde{s}_i|.$$

Based on clinician ratings, we define both $MSE_{clinician}$ and $MAE_{clinician}$ in a similar fashion. These metrics are used to compare individual crowdsourced scores, clinician scores, and AI-based scores.

## Convolutional neural network

For the automated scoring of ROCF drawings, two variations of deep learning systems were implemented: a regression-based network and a multilabel classification network. The multilabel classification problem is based on the assumption that one drawing can contain multiple classes. That is, the scores for a specific element correspond to four mutually exclusive classes (i.e. 0, 0.5, 1, and 2), which corresponds to the original Osterrieth scoring system (**Meyers et al., 1996**; see also **Figure 1— figure supplement 1**). Each class can appear simultaneously with score classes corresponding to other elements in the figure. Although the scores for different elements share common features, such as texture or drawing style, which are relevant to the element score, the scoring of each element can be viewed as an independent process, taking these shared features into account. These considerations are the starting point for the architectural design of our scoring system, shown in **Figure 2A**. The architecture consists of a shared feature extractor network and 18 parallel decision heads, one for each item score. The shared feature extractor is composed of three blocks, introducing several model hyperparameters: Each block consists of a 3 × 3 convolutional layer, batch normalization, 3 × 3 convolutional layer, batch normalization and max pooling with 2 × 2 stride. The ReLU activation function is applied after each batch normalization and dropout is applied after max pooling. The output of each block is a representation with 64, 128, and 256 channels, respectively. Each of the 18 per-item networks is applied on this representation. These consist of a similar feature extractor block, but with global max pooling instead, which outputs a 512-dimensional representation that is subsequently fed into a fully connected layer, that outputs a 1024-dimensional vector, followed by batch normalization, ReLU, dropout, and an output fully connected layer.

Two variations of this network were implemented: a regression-based network and a multilabel classification network. They differ in the output layer and in the loss function used during training. The regression-based network outputs a single number, while the multilabel classification network outputs class probabilities, each of which corresponds to one of the four different item scores {0, 0.5, 1.0, 2.0}. Second, the choice of loss functions also incurs different dynamics during optimization. The MSE loss used for the regression model penalizes smaller errors less than big errors, for example, for an item with a score 0.5, predicting the score 1.0 is penalized less than when the model predicts 2.0. In other words, the MSE loss naturally respects the ordering in the output space. This is in contrast to the multilabel classification model for which a cross entropy loss was used for each item, which does not differentiate between different classes in terms of 'how wrong' the model is: in the example above, 1.0 is considered equally wrong as 2.0.

## Data augmentation

In a further study, we investigated the effect of DA during training from two perspectives. First, DA is a standard technique to prevent machine-learning models from overfitting to training data. The intuition is that enriching the dataset with perturbed versions of the original images leads to better generalization by enriching the dataset with an additional and more diverse training set. Second, DA can also help in making models more robust against semantic transformations like rotations or changes in perspective. These transformations are particularly relevant for the present application

since users in real-life are likely to take pictures of drawings which might be slightly rotated or with a slightly tilted perspective. With these intuitions in mind, we randomly transformed drawings during training. Each transformation was a combination of Gaussian blur, a random perspective change and a rotation with angles chosen randomly between −10° and 10°. The DA did not include generative models. Initially, we explored using generative models, specifically generative adversarial networks (GANs), for DA to address the scarcity of low-score images compared to high-score images. However, due to the extensive available dataset, we did not observe any substantial performance improvements in our model. Nevertheless, future studies could explore generative models, such as variational auto-encoders or Bayesian networks, which can then be tested on the data from the current prospective study and compared with our results.

## Training and validation

To evaluate our model, we set aside 4045 of the 20,225 ROCF drawings as a testing set, corresponding to 20% of the dataset. The remaining 16,180 images were used for training and validation. Specifically, we used 12,944 (80%) drawings for training and 3236 (20%) as a validation set for early stopping.

The networks and training routines were implemented in PyTorch 1.8.1 (*Paszke et al., 2019*). The training procedure introduces additional hyperparameters: Our main model was trained on drawings of size 232 × 300 (see *Figure 2D* and below on details on the image resolution analysis), using batches of 16 images. Model parameters were optimized by the Adam optimizer (*Kingma and Ba, 2014*) with the default parameters $\beta 1 = 0.9$, $\beta 2 = 0.999$, and $\varepsilon = 1e-8$. The initial learning rate, set to 0.01, was chosen to facilitate stable model convergence. It decayed exponentially by a factor of 0.95 per epoch, gradually reducing the step size to prevent overshooting the optimal solution during training. To prevent overfitting dropout rates were set to 0.3 and 0.5 in the convolutional and fully connected layers, respectively. Additionally, we used the validation split (20%) of the training data for early stopping. Since our dataset is imbalanced and contains a disproportionately high number of high-score drawings, we sampled the drawings in a way that resulted in evenly distributed scores in a single batch. We trained the model for 75 epochs and picked the weights corresponding to the epoch with the smallest validation loss. The multilabel classification model was trained with cross entropy loss applied to each item classifier independently so that the total loss is obtained by summing the individual loss terms. The regression model was trained in an analogous manner except that we used the MSE for each item score, instead of the cross entropy loss.

## Performance evaluation

We evaluated the overall performance of the final model on the test set based on MSE, MAE, and *R*-squared. Even though item-specific scores are discrete and we also implement a classification-based approach, we chose not to rely on accuracy to assess performance. Because accuracy penalizes errors equally, minor differences in scores can lead to arbitrarily small accuracy, or even worse scores than a model that leads to bigger differences on average. MAE gives a better picture of how accurate the model is.

Performance metrics (MSE, MAE, and *R*-squared) were examined for the total score and additionally for each of the 18 items individually. To probe the performance of the model across a wide range of scores, we also obtained MSE and MAE for each total score. Given predictions $\hat{y}$, true scores $y$, and the average of true scores $\bar{y}$, MSE, MAE, and *R*-squared are defined, respectively, as

$$MSE = \frac{1}{N} \sum_{i}^{N} \left( y_i - \hat{y}_i \right)^2,$$

$$MAE = \frac{1}{N} \sum_{i}^{N} |y_i - \hat{y}_i|, \text{ and}$$

$$R^2 = 1 - \sum_{i}^{N} \left( y_i - \hat{y}_i \right)^2 / \sum_{i}^{N} \left( y_i - \bar{y} \right)^2.$$

In addition, we evaluated the performance in a pre-registered (https://osf.io/82796) independent prospective validation study. Importantly, both datasets (original test set i.e. 4045 images) and the independent prospective dataset (i.e. 2498 images) were never seen by the neural networks.

### Test-time augmentation

TTA was performed to promote robustness at test-time to slight variations in input data. First, we experimented with the same augmentations that we used in our DA training policy, namely randomly applying the perspective change, resizing the image, and applying rotations. In addition to the original image, we fed four to seven randomly transformed drawings (see description of DA for the possible transformations) to the model and averaged the output logits to obtain a prediction. In addition, we experimented with approaches exclusively applying either random rotations or random perspective change. None of these random procedures improved the model performance. Nonetheless, performance improvement was achieved by applying deterministic rotations along positive and negative angles on the unit circle. We performed five rotations, uniformly distributed between −2° and 2°. For this choice, computational restrictions for the model's intended use case were considered. Specifically, time complexity increases linearly with the number of augmentations. In keeping a small set of augmentations, we believe that small perturbations to rotation present an obvious correspondence to the variations the model will be exposed to in a real application.

### Final model selection

Both the model and training procedure contain various hyperparameters and even optional building blocks such as using DA during training, as well as applying TTA during inference. After optimizing the hyperparameters of each model independently, all of the possible variants that emerge from applying DA and TTA were explored on both classification and regression models. Therefore, the space of possible model variants is spanned by the non-augmented model, the model trained with DA, the model applying TTA during inference, as well as the model variant applying both aforementioned building blocks.

To take advantage of particularities in both models, a mixture of the best performing regression and classification models was obtained. In this model, the per-item performances of both models are compared on the held-out validation set. During inference, for each item, we output the prediction of the best performing model. Therefore, on a figure-wide scale, the prediction of the combined model uses both classification and regression models concurrently, while not combining the models' per-item predictions.

### Robustness analysis

We investigated the robustness of our model against different semantic transformations which are likely to occur in real-life scenarios. These were rotations, perspective changes, and changes to brightness and contrast. To assess the robustness against these transformations, we transformed images from our test set with different transformation parameters. For rotations, we rotated drawings with angles chosen randomly from increasingly higher orders, both clockwise and counterclockwise. That is, we sampled angles between 0° and 5°, between 5° and 10°, and so on, up to 40° to 45°. The second transform we investigated were changes in perspective as these are also likely to occur, for example when photos of drawings are taken with a camera in a tilted position. The degree of perspective change was guided by a parameter between 0 and 1, where 0 corresponds to the original image and 1 corresponds to an extreme change in perspective, making the image almost completely unrecognizable. Furthermore, due to changes in light conditions, an image might appear brighter or with a different contrast. We used brightness parameters between 0.1 and 2.0, where values below 1.0 correspond to a darkening of the image and values above 1.0 correspond to a brighter image. Similarly, for contrast, we varied the contrast parameter between 0.1 and 2.0, with values above (below) 1.0 corresponding to high (low) contrast images.

### Image resolution analysis

The ROCF images in our dataset have varying resolutions, ranging from 100 × 140 pixels to 3500 × 5300 pixels. Since our models are trained on a fixed input resolution, we investigated the effect of different resolutions on the model performance measured in terms of MAE. To that end, we trained the

multilabel classification model with and without DA for inputs of size 78 × 100, 116 × 150, 232 × 300, and 348 × 450. In principle, when using smaller images, then more information is lost due to resizing, while, on the other hand, a resolution which is too large requires bigger models due to increased complexity in the underlying distribution. In addition, using our DA pipeline with small images might negatively affect the performance since the interpolation techniques used in the semantic transformations like rotations, potentially leads to an additional loss of information. We observed this in our experiments which showed that inputs of size 232 × 300 yielded the best performance, both for the model with and without DA (*Figure 2E*). Thus, all subsequent analyses were performed with images of size 232 × 300.

## Acknowledgements

This work was supported by the URPP 'Dynamics of Healthy Aging' and BRIDGE [40B2-0_187132], which is a joint programme of the Swiss National Science Foundation SNSF and Innosuisse. Furthermore, BHV is funded by the Swiss National Science Foundation [10001C_197480]. Finally, CL is supported by the Swiss State Secretariat for Education, Research and Innovation (SERI) under contract number MB22.00036. None of the authors have been paid to write this article by a pharmaceutical company or other agency. Authors were not precluded from accessing data in the study, and they accept responsibility to submit for publication.

## Additional information

### Funding

| Funder | Grant reference number | Author |
| --- | --- | --- |
| URPP "Dynamics of Healthy Aging" | | Nicolas Langer |
| Swiss National Science Foundation | BRIDGE 40B2-0_187132 | Nicolas Langer<br>Ce Zhang |
| Swiss National Science Foundation | 10001C_197480 | Nicolas Langer<br>Bruno Hebling Vieira |
| State Secretariat for Education, Research and Innovation | MB22.00036 | Ce Zhang |

The funders had no role in study design, data collection, and interpretation, or the decision to submit the work for publication.

### Author contributions

Nicolas Langer, Conceptualization, Resources, Software, Formal analysis, Supervision, Visualization, Methodology, Writing - original draft, Project administration, Writing – review and editing; Maurice Weber, Conceptualization, Software, Formal analysis, Validation, Visualization, Methodology, Writing - original draft, Project administration, Writing – review and editing; Bruno Hebling Vieira, Lukas Wolf, Conceptualization, Data curation, Software, Formal analysis, Validation, Visualization, Methodology, Writing - original draft, Writing – review and editing; Dawid Strzelczyk, Conceptualization, Resources, Data curation, Software, Formal analysis, Validation, Visualization, Methodology, Writing - original draft, Writing – review and editing; Andreas Pedroni, Jonathan Heitz, Stephan Müller, Christoph Schultheiss, Conceptualization, Methodology, Writing – review and editing; Marius Troendle, Conceptualization, Resources, Data curation, Methodology, Writing – review and editing; Juan Carlos Arango Lasprilla, Diego Rivera, Federica Scarpina, Qianhua Zhao, Resources, Data curation, Writing – review and editing; Rico Leuthold, Conceptualization, Software, Methodology; Flavia Wehrle, Oskar Jenni, Peter Brugger, Tino Zaehle, Conceptualization, Resources, Writing – review and editing; Romy Lorenz, Conceptualization, Methodology, Writing - original draft, Writing – review and editing; Ce Zhang, Conceptualization, Resources, Formal analysis, Supervision, Funding acquisition, Methodology, Writing – review and editing

## Author ORCIDs
Nicolas Langer (ID) https://orcid.org/0000-0002-6038-9471
Bruno Hebling Vieira (ID) https://orcid.org/0000-0002-8770-7396
Diego Rivera (ID) https://orcid.org/0000-0001-7477-1893
Tino Zaehle (ID) https://orcid.org/0000-0003-3673-4869

## Ethics
The study was approved by the Institutional Ethics Review Board of the 'Kantonale Ethikkommission' (BASEC-Nr. 2020-00206). All collaborators have written informed consent and/or data usage agreements for the recorded drawings from the participants. The authors assert that all procedures contributing to this work comply with the ethical standards of the relevant national and institutional committees on human experimentation and with the Helsinki Declaration of 1975, as revised in 2008.

Reviewer #1 (Public review): https://doi.org/10.7554/eLife.96017.3.sa1
Reviewer #2 (Public review): https://doi.org/10.7554/eLife.96017.3.sa2
Reviewer #3 (Public review): https://doi.org/10.7554/eLife.96017.3.sa3
Author response https://doi.org/10.7554/eLife.96017.3.sa4

# Additional files

## Supplementary files
• MDAR checklist

## Data availability
The clinical dataset cannot be shared publicly due to the absence of consent from patients for data sharing. The Prolific dataset, deidentified raw data (i.e. image of the ROCF) can be accessed through OSF (https://osf.io/uea6f). Processed data used in analyses, such as summary statistics and numbers used to plot figures in the manuscript, are available as source data. All preprocessing and analysis scripts used in this study are available on GitHub (https://github.com/methlabUZH/rey-figure; copy archived at *Weber, 2024*). Researchers interested in accessing any data or materials should contact the corresponding author for further instructions and to discuss the appropriate access procedures.

The following dataset was generated:

| Author(s) | Year | Dataset title | Dataset URL | Database and Identifier |
|---|---|---|---|---|
| Langer N, Strzelczyk D | 2024 | Open Science Framework | https://osf.io/uea6f | Open Science Framework, uea6f |

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
