## [Editor Report · eLife assessment]

The methods and findings of the current work are **important** and well-grounded. The strength of the evidence presented is **convincing** and backed up by rigorous methodology. The work, when elaborated on how to access the app, will have far-reaching implications for current clinical practice.

---

## [Referee Report · Reviewer #1 (Public review)]

Summary:

The authors aimed to develop and validate an automated, deep learning-based system for scoring the Rey-Osterrieth Complex Figure Test (ROCF), a widely used tool in neuropsychology for assessing memory deficits. Their goal was to overcome the limitations of manual scoring, such as subjectivity and time consumption, by creating a model that provides automatic, accurate, objective, and efficient assessments of memory deterioration in individuals with various neurological and psychiatric conditions.

Strengths:

Comprehensive Data Collection: The authors collected over 20,000 hand-drawn ROCF images from a wide demographic and geographic range, ensuring a robust and diverse dataset. This extensive data collection is critical for training a generalizable and effective deep learning model.

Advanced Deep Learning Approach: Utilizing a multi-head convolutional neural network to automate ROCF scoring represents a sophisticated application of current AI technologies. This approach allows for detailed analysis of individual figure elements, potentially increasing the accuracy and reliability of assessments.

Validation and Performance Assessment: The model's performance was rigorously evaluated against crowdsourced human intelligence and professional clinician scores, demonstrating its ability to outperform both groups. The inclusion of an independent prospective validation study further strengthens the credibility of the results.

Robustness Analysis Efficacy: The model underwent a thorough robustness analysis, testing its adaptability to variations in rotation, perspective, brightness, and contrast. Such meticulous examination ensures the model's consistent performance across different clinical imaging scenarios, significantly bolstering its utility for real-world applications.

Appraisal and discussion:

By leveraging a comprehensive dataset and employing advanced deep learning techniques, they demonstrated the model's ability to outperform both crowdsourced raters and professional clinicians in scoring the ROCF. This achievement represents a significant step forward in automating neuropsychological assessments, potentially revolutionizing how memory deficits are evaluated in clinical settings. Furthermore, the application of deep learning to clinical neuropsychology opens avenues for future research, including the potential automation of other neuropsychological tests and the integration of AI tools into clinical practice. The success of this project may encourage further exploration into how AI can be leveraged to improve diagnostic accuracy and efficiency in healthcare.

However, the critique regarding the lack of detailed analysis across different patient demographics, the inadequacy of network explainability, and concerns about the selection of median crowdsourced scores as ground truth raises questions about the completeness of their objectives. These aspects suggest that while the aims were achieved to a considerable extent, there are areas of improvement that could make the results more robust and the conclusions stronger.

Comments on revised version:

I appreciate the opportunity to review this revised submission. Having considered the other reviews, I believe this study presents an important advance in using AI methods for clinical applications, which is both innovative and has implications beyond a single subfield.

The authors have developed a system using fundamental AI that appears sufficient for clinical use in scoring the Rey-Osterrieth Complex Figure (ROCF) test. In human neuropsychology, tests that generate scores like this are a key part of assessing patients. The evidence supporting the validity of the AI scoring system is compelling. This represents a valuable step towards evaluating more complex neurobehavioral functions.

However, one area where the study could be strengthened is in the explainability of the AI methods used. To ensure the scores are fully transparent and consistent for clinical use, it will be important for future work to test the robustness of the approach, potentially by comparing multiple methods. Examining other latent variables that can explain patients' cognitive functioning would also be informative.

In summary, I believe this study provides an important proof-of-concept with compelling evidence, while also highlighting key areas for further development as this technology moves towards real-world clinical applications.

---

## [Referee Report · Reviewer #2 (Public review)]

The authors aimed to develop and validate a machine-learning driven neural network capable of automatic scoring of the Rey-Osterrieth Complex Figure. They aimed to further assess the robustness of the model to various parameters such as tilt and perspective shift in real drawings. The authors leveraged the use of a huge sample of lay workers in scoring figures and also a large sample of trained clinicians to score a subsample of figures. Overall, the authors found their model to have exceptional accuracy and perform similarly to crowdsourced workers and clinicians with, in some cases, less degree of error/score dispersion than clinicians.

---

## [Referee Report · Reviewer #3 (Public review)]

This study presented a valuable inventory of scoring a neuropsychological test, ROCFT, with constructing an artificial intelligence model.

Comments on latest version:

The authors made the system with fundamental AI that is sufficient for clinical use for humans. In human neuropsychology, the test that generates the score is fundamental and relatively easy. Neuropsychologists apply patients to many tests; therefore, the present system is one of them, where we cannot tell the total neurofunction of a patient. The evidence for scoring is thought to be compelling quality, enough for clinical use now and we progress to evaluate other more complicated human neuropsychological functions. For example, persons with dementia change their performance easily when they feel other emotions (worry, boredom, etc.) and notice other stimulation (announcements in the hospital, a walking nurse by chance, etc.). The score of ROCF is definitely changing, compelling the effort of AI scoring. We should grasp this behavior of humans with diverse tests totally. Therefore, scoring AI with compelling quality is a fundamental step for the next, evaluation against the changeable and ambiguous neurobehavior of humans.

---

## [Author Response]

The following is the authors’ response to the original reviews.

**Reviewer #1:**
Comment #1: Insufficient Network Analysis for Explainability: The paper does not sufficiently delve into network analysis to determine whether the model's predictions are based on accurately identifying and matching the 18 items of the ROCF or if they rely on global, item-irrelevant features. This gap in analysis limits our understanding of the model's decision-making process and its clinical relevance.

Response #1: Thank you for your comment. We acknowledge the importance of understanding the decision-making process of AI models is crucial for their acceptance and utility in clinical settings. However, we believe that our current approach, which focuses on providing individual scores for each of the 18 items of the Rey-Osterrieth Complex Figure (ROCF), inherently offers a higher level of explainability and practical utility for clinicians than a network analysis could. Our multi-head convolutional neural network is designed with a dedicated output head for each of the 18 items in the ROCF, and thus provides separate scores for each of the 18 items in the ROCF. This architecture helps that the model focuses on individual elements rather than relying on global, item-irrelevant features.

This item-specific approach directly aligns with the traditional clinical assessment method, thereby making the results more interpretable and actionable for clinicians. The individual scores for each item provide detailed insights into a patient's performance. Clinicians can use these scores to identify specific areas of strength and weakness in a patient's visuospatial memory and drawing abilities.

Furthermore, we evaluated the model's performance on each of the 18 items separately, providing detailed metrics that show consistent accuracy across all items. This item-level performance analysis offers clear evidence that the model is not relying on irrelevant global features but is indeed making decisions based on the specific characteristics of each item. We believe that our approach provides a level of explainability that is directly useful and relevant to clinical practitioners.

Comment #2: Generative Model Consideration: The critique suggests exploring generative models to model the joint distribution of images and scores, which could offer deeper insights into the relationship between scores and specific visual-spatial disabilities. The absence of this consideration in the study is seen as a missed opportunity to enhance the model's explainability and clinical utility.

Response #2: Thank you for your thoughtful comment and the suggestion to explore generative models. We appreciate the potential benefits that generative models to model the joint distribution of images and scores. However, we chose not to pursue this approach in our study for several reasons: First, our primary goal was to develop a model that provides accurate and interpretable scores for each of the 18 individual items in the ROCF figure. Second, generative models, while powerful, would add a layer of complexity that might diminish the clarity and immediate clinical applicability of our results. Generative models, (particularly deep learning-based) can be challenging to interpret in terms of how they make decisions or why they produce specific outputs. This lack can be a concern in critical applications involving neurological and psychiatric disorders. Clinicians require tools that provide clear insights without the need for additional layers of analysis. Our current model provides detailed, item-specific scores that clinicians can directly use to assess visuospatial memory and drawing abilities. Initially, we explored using generative models (i.e. GANs) for data augmentation to address the scarcity of low-score images compared to high-score images. Moreover, for the low-score images, the same score can be achieved by numerous combinations of figure elements. However, due to our extensive available dataset, we did not observe any substantial performance improvements in our model. Nevertheless, future studies could explore generative models, such as Variational Autoencoders (VAEs) or Bayesian Networks, and test them on the data from the current prospective study to compare their performance with our results.

In the revised manuscript, we have included additional sentences discussing the potential use of generative models and their implications for future research.

“The data augmentation did not include generative models. Initially, we explored using generative models, specifically GANs, for data augmentation to address the scarcity of low-score images compared to high-score images. However, due to the extensive available dataset, we did not observe any substantial performance improvements in our model. Nevertheless, Future studies could explore generative models, such as Variational Autoencoders (VAEs) or Bayesian Networks, which can then be tested on the data from the current prospective study and compared with our results.”

Comment #3: Lack of Detailed Model Performance Analysis Across Subject Conditions: The study does not provide a detailed analysis of the model's performance across different ages, health conditions, etc. This omission raises questions about the model's applicability to diverse patient populations and whether separate models are needed for different subject types.

Response #3: Thank you for your this important comment. Although the initial version of our manuscript already provided detailed “item-specific” and “across total scores” performance metrics, we recognize the importance of including detailed analyses across different patient demographics to enhance the robustness and applicability of our findings. In response to your comment, we have conducted additional analyses that provide a comprehensive evaluation of model performance across various patient demographics, such as age groups, gender, and different neurological and psychiatric conditions. This additional analysis demonstrates the generalizability and reliability of our model across diverse populations. We have included these analyses in the revised manuscript.

“In addition, we have conducted a comprehensive model performance analysis to evaluate our model's performance across different ROCF conditions (copy and recall), demographics (age, gender), and clinical statuses (healthy individuals and patients) (Figure 4A). These results have been confirmed in the prospective validation study (Supplementary Figure S6). Furthermore, we included an additional analysis focusing on specific diagnoses to assess the model's performance in diverse patient populations (Figure 4B). Our findings demonstrate that the model maintains high accuracy and generalizes well across various demographics and clinical conditions.”

Comment #4: Data Augmentation: While the data augmentation procedure is noted as clever, it does not fully encompass all affine transformations, potentially limiting the model's robustness.

Response #4: We appreciate your feedback on our data augmentation strategy. We acknowledge that while our current approach significantly improves robustness against certain semantic transformations, it may not fully cover all possible affine transformations.

Here, we provide further clarification and justification for our chosen methods and their impact on the model's performance: In our study, we implemented a data augmentation pipeline to enhance the robustness of our model against common and realisitc geometric and semantic-preserving transformations. This pipeline included rotations, perspective changes, and Gaussian blur, which we found to be particularly effective in improving the model's resilience to variations in input data. These transformations are particularly relevant for the present application since users in real-life are likely to take pictures of drawings that might be slightly rotated or with a slightly tilted perspective. With these intuitions in mind, we randomly transformed drawings during training. Each transformation was a combination of Gaussian blur, a random perspective change, and a rotation with angles chosen randomly between -10° and 10°. These transformations are representative of realistic scenarios where images might be slightly tilted or photographed from different angles. We intentionally did not explicitly address all affine transformations, such as shearing or more complex geometric transformations because these transformations could alter the score of individual items of the ROCF and would be disruptive to the model.

As noted in our manuscript and demonstrated in supplementary Figure S1, the data augmentation pipeline significantly improved the model's robustness against rotations and changes in perspective. Importantly, our tablet-based scoring application can further ensure that the photos taken do not exhibit excessive semantic transformations. By leveraging the gyroscope built into the tablet, the application can help users align the images properly, minimizing issues such as excessive rotation or skew. This built-in functionality helps maintain the quality and consistency of the images, reducing the likelihood of significant semantic transformations that could affect model performance.

Comment #5: Additionally, the rationale for using median crowdsourced scores as ground truth, despite evidence of potential bias compared to clinician scores, is not adequately justified.

Response #5: Thank you for this valuable comment. Clarifying the rationale behind using the median score of crowdsourcing as the ground truth is indeed important. To reach high accuracy in predicting individual sample scores of the ROCFs, it is imperative that the scores of the training set are based on a systematic scheme with as little human bias as possible influencing the score. However, our analysis (see results section) and previous work (Canham et al., 2000) suggested that the scoring conducted by clinicians may not be consistent, because the clinicians may be unwittingly influenced by the interaction with the patient/participant or by the clinicians factor (e.g. motivation and fatigue). For this reason and the incomplete availability of clinician scores for all figures (i.e. for 19% of the 20’225 figures), we did not use the clinicians scores as ground truth scores. Instead, we have trained a large pool (5000 workers) of human internet workers (crowdsourcing) to score ROCFs drawings guided by our self-developed interactive web application. Each element of the figure was scored by several human workers (13 workers on average per figure). We have obtained the ground truth for each drawing by computing the median for each item in the figure, and then summed up the medians to get the total score for the drawing in question. To further ensure high-quality data annotation, we identified and excluded crowdsourcing participants that have a high level of disagreement (>20% disagreement) with this rating from trained clinicians, who carefully scored manually a subset of the data in the same interactive web application.

We chose the median score for several reasons: (1) the median score is less influenced by outliers compared to the mean. Given the variability of scoring between different clinicians and human workers (see human MSE and clinician MSE), using the median ensures that the ground truth is not skewed by extreme values, leading to more stable and reliable scores. (2) Crowdsource data do not always follow a normal distribution. In cases where the distribution is skewed or not symmetric, the median can be a more representative measure of the center. (3) The original scoring system involves ordinal scales (0,0.5,1,2). For ordinal scales, the median is often more appropriate than the mean. Finally, by aggregating multiple scores from a large pool of crowdsourced raters, the median provides a consensus that reflects the most common assessment. This approach mitigates the variability introduced by individual rater biases and ensures a more consistent ground truth. In clinical settings, the consensus of multiple expert opinions often serves as the benchmark for assessments. The use of median scores mirrors this practice, providing a ground truth that is representative of collective human judgment.

Canham, R. O., S. L. Smith, and A. M. Tyrrell. 2000. “Automated Scoring of a Neuropsychological Test:

The Rey Osterrieth Complex Figure.” Proceedings of the 26th Euromicro Conference. EUROMICRO 2000. Informatics: Inventing the Future. https://doi.org/10.1109/eurmic.2000.874519.

**Reviewer #2:**
Comment #1: There is no detail on how the final scoring app can be accessed and whether it is medical device-regulated.

Response #1: We appreciate the opportunity to provide more information about the current status and plans for our scoring application. At this stage, the final scoring app is not publicly accessible as it is currently undergoing rigorous beta testing with a select group of clinicians in real-world settings. The feedback from these clinicians is instrumental in refining the app’s features, interface, and overall functionality to improve its usability and user experience. This ensures that the app meets the high standards required for clinical tools. Following the successful completion of the beta testing phase, we aim to seek FDA approval for the scoring app. Achieving this regulatory milestone will guarantee that the app meets the stringent requirements for medical devices, providing an additional layer of confidence in its safety and efficacy for clinical use. Once FDA approval is obtained, we plan to make the app publicly accessible to clinicians and healthcare institutions worldwide. Detailed instructions on how to access and use the app will be provided at that time on our website (https://www.psychology.uzh.ch/en/areas/nec/plafor/research/rfp.html).

Comment #2: No discussion on the difference in sample sizes between the pre-registration of the prospective study and the results (e.g., aimed for 500 neurological patients but reported data from 288). Demographics for the assessment of the representation of healthy and non-healthy participants were not present.

Response #2: Thank you for your comment. We believe there might have been a misunderstanding regarding our preregistration details. In the preregistration, we planned to prospectively acquire ROCF drawings from 1000 healthy subjects. Each subject should have drawn two ROCF drawings (copy and memory condition). Consequently, 2000 samples should have been collected. In addition, in our pre-registration plan, we aimed to collect 500 drawings from patients (i.e. 250 patients), not 500 patients as the reviewer suggested (https://osf.io/82796). Thus in total, the goal was to obtain 2500 ROCF figures. The final prospective data set, which contained 2498 ROCF images from 961 healthy adults and 288 patients very closely matches the sample size, we aimed for in the the pre-registration. We do not see a necessity to discuss this negligible discrepancy in the main manuscript. The prospective data set remains substantial and sufficient to test our model on the independent prospective data set. Importantly, we want to highlight that the test set in the retrospective data set (4045 figures) was also never seen by the model. Both the retrospective and prospective data sets demonstrate substantial global diversity as the data has been collected in 90 different countries. Please note, that Supplementary Figures S2 & S3 provide detailed demographics of the participants in the prospectively collected data, present their performance in the copy and (immediate) recall condition across the lifespan, and the worldwide distribution of the origin of the data.

Comment #3: Supplementary Figure S1 & S4 is poor quality, please increase resolution.

Response #3: We apologize for the poor quality of Supplementary Figures S1 and S4 in the initial submission. In the revised version of our submission, we have increased the resolution of both Supplementary Figure S1 and Supplementary Figure S4 to ensure that all details are clearly visible and the figures are of high quality.

Comment #4: Regarding medical device regulation; if the app is to be used in clinical practice (as it generates a score and classification of performance), I believe such regulation is necessary - but there are ways around it. This should be detailed.

Response #4: We agree that regulation is essential for any application intended for use in clinical practice, particularly one that generates scores and classifications of performance. As discussed in response #1, the final scoring application is currently undergoing intensive beta testing in real-world settings with a limited group of clinicians and is therefore not publicly accessible at this time. We are fully committed to obtaining the necessary regulatory approvals before the app is made publicly accessible for clinical use. Once the beta testing phase is complete and the app has been refined based on clinician feedback, we will prepare and submit a comprehensive regulatory dossier. This submission will include all necessary data on the app's development, testing, validation, and clinical utility. We are adhering to relevant regulatory standards and guidelines, such as ISO 13485 for medical devices and the FDA's guidance on software as a medical device (SaMD).

Comment #7: Need to clarify that work was already done and pre-printed in 2022 for the main part of this study, and that this paper contributes to an additional prospective study.

Response #7: We would like to clarify that the pre-print the reviewer is referring to is indeed the current paper submitted to ELife. The submitted paper includes both the work that was pre-printed in 2022 and the additional prospective study, as you correctly identified.

**Reviewer #3:**
Comment #1: The considerable effort and cost to make the model only for an existing neuropsychological test.

Response #1: We acknowledge that significant effort and resources were dedicated to developing our model for the Rey-Osterrieth Complex Figure (ROCF) test. Below, we provide a detailed rationale for this investment and the broader implications of our work. The ROCF test is one of the most widely used neuropsychological assessments worldwide, providing critical insights into visuospatial memory and executive function. While the initial effort and cost are substantial, the long-term benefits of an automated, reliable, objective, fast and widely applicable neuropsychological assessment tool justify the investment. The scoring application will significantly reduce the time for scoring the test and thus provide more efficient use of clinical resources, and the potential for broader applications makes this a worthwhile endeavor. The methods and infrastructure developed for this model can be adapted and scaled to other neuropsychological tests and assessments (e.g. Taylor Figure).

Comment #2: I was truly impressed by the authors' establishment of a system that organizes the methods and fields of diverse specialties in such a remarkable way. I know the primary purpose of ROCFT. However, beyond the score, neuropsychologically, these are observed by specialists while ROCFT and that is attractive of the test: the turn of each stroke (e.g., from right to left, from the main structure to the margin or small structure), the process to total completeness as a figure, e.g., confidential speed and concentration, the boldness of strokes, unnatural fragmentation of strokes, the not deviated place in a paper, turning of the figure itself (before the scanning level), the total size, the level compared with the age, education, and experiences of the patient. Those are reflected by the disease, visuospatial intelligence, executive function, and ability to concentrate. Scores are crucial, but by observing the drawing process, we can obtain diverse facts or parts of symptoms that imply the complications of human behavior.

Response #2: Thank you for your insightful comments and observations regarding our system for organizing diverse specialties within the ROCFT methodology. We agree that beyond the numerical scores, the detailed observation of the drawing process provides invaluable neuropsychological insights. How strokes are executed, from their direction and placement to the overall completion process, offers a nuanced understanding of factors like spatial orientation, concentration, and executive function. In fact, we are working on a ROCF pen tracking application, which enables the patient to draw the ROCF with a digital pen on a tablet. The tablet can (1) assess the sequence order of drawing the items and the number of strokes, (2) record the exact coordinate of each drawn pixel at each time point of the assessment, (3) measure the duration for each pen stroke as well as total drawing time, and (4) assess the pen stroke pressure. Through this, we aim to extract additional information on processing speed, concentration, and other cognitive domains. However, this development is outside the scope of the current manuscript.